# Performance Comparison of Five Methods Available in ImageJ for Bird Counting and Detection from Video Datasets

**Kevin Adi Kurnia** [1,2], **Ferry Saputra** [1,2], **Cao Thang Luong** [3], **Marri Jmelou M. Roldan** [4], **Tai-Sheng Cheng** [5,*] and **Chung-Der Hsiao** [1,2,6,7,*]

1   Department of Chemistry, Chung Yuan Christian University, Chung-Li 32023, Taiwan; kevinadik-adi@hotmail.com (K.A.K.); ferrysaputratj@gmail.com (F.S.)
2   Department of Bioscience Technology, Chung Yuan Christian University, Chung-Li 32023, Taiwan
3   Department of Chemical Engineering & Institute of Biotechnology and Chemical Engineering, I-Shou University, Da-Shu, Kaohsiung City 84001, Taiwan; thang.luongcao@gmail.com
4   Faculty of Pharmacy, The Graduate School, University of Santo Tomas, Manila 1008, Philippines; mmroldan@ust.edu.ph
5   Department of Biological Sciences and Technology, National University of Tainan, Tainan 70005, Taiwan
6   Center for Nanotechnology, Chung Yuan Christian University, Chung-Li 32023, Taiwan
7   Research Center for Aquatic Toxicology and Pharmacology, Chung Yuan Christian University, Chung-Li 32023, Taiwan
*   Correspondence: taishengcheng@yahoo.com.tw (T.-S.C.); cdhsiao@cycu.edu.tw (C.-D.H.)

**Abstract:** Bird monitoring is an important approach to studying the diversity and abundance of birds, especially during migration, as it can provide core data for bird conservation purposes. The previous methods for bird number estimation are largely based on manual counting, which suffers from low throughput and a high error rate. In this study, we aimed to provide an alternative bird-counting method from video datasets by using five available ImageJ methods: Particle Analyzer, Find Maxima, Watershed segmentation, TrackMate, and trainable WEKA segmentation. The numbers of birds and their XY coordinates were extracted from videos to conduct a side-by-side comparison with the manual counting results, and the three important criteria of the sensitivity, precision, and F1 score were calculated for the performance evaluation. From the tests, which we conducted for four different cases with different bird numbers or flying patterns, TrackMate had the best overall performance for counting birds and pinpointing their locations, followed by Particle Analyzer, Find Maxima, WEKA, and lastly, Watershed, which showed low precision in most of the cases. In summary, five ImageJ-based counting methods were compared in this study, and we validated that TrackMate obtains the best performance for bird counting and detection.

**Keywords:** bird counting; ImageJ; TrackMate; video dataset

## 1. Introduction

In recent years, habitat loss, climate change, and human activity have begun to pose a great threat to biodiversity [1]. Birds are one example of the most impacted animals due to these external conditions, which alter their migratory behavior, thereby changing the time, distance, and routes of their migration, which might be challenging to the birds [2]. Monitoring wildlife activity, especially observing and studying the populations and movements of birds, is important in the advancement of scientific understanding and ecological conservation efforts, as birds have proven to be essential markers of the health of ecosystems due to their richness and distinctive ecological role as biodiversity indicators [1,3–5].

A practical and affordable real-time approach to identifying environmental changes is through bird monitoring [1,5,6]. The information on changes in migration patterns [7], breeding behavior changes [8], and population variations through systematic monitoring enables scientists and conservationists to react swiftly to new environmental issues. Scientists use various techniques to observe birds' movements, from simple visual observation

to phone cameras, digital cameras, drones, and even satellite imaging. These methods can be used separately or in tandem, depending on the objectives of the study, the size of the research area, and the characteristics of the bird species being observed [9]. Additionally, with the advancements in technology and the creation of scientific platforms, birdwatchers can now submit their observations to international databases [1,10,11]. This collection of data can be used to investigate distribution patterns [12], population dynamics [7,13], and the impact of environmental change on bird communities [14] (Table 1).

Observing their movement patterns, including their migration, nomadism, dispersal, altitudinal movement, weather-related movement, and daily movement patterns, is key to the design of effective strategies to conserve their populations. For example, *Sterna paradisaea* migrates from the Arctic breeding ground to the Antarctic region following a specific route triggered by season change and the availability of food supplies along their route [15]. Another example is *Selasphorus rufus*, which shows altitudinal migration during the breeding season to follow the blooming of flowers that fit their preference [16]. Another popular bird movement shown by starlings is called murmuration. During murmurations, large flocks of starlings move in coordinated patterns and show mesmerizing aerial displays. Environmental factors such as avoiding predators and the location of the roosting site also affect the movement of the patterns [10].

**Table 1.** Bird-monitoring methods that are currently utilized in practice.

| Tools | Description | Advantage | References |
|---|---|---|---|
| eBird and citizen science platforms | Online platforms like eBird allow bird watchers to submit checklists of bird observations, contributing to large-scale databases. | Massive data collection, global coverage, engagement of citizen scientists. | [1,10,11] |
| Transect surveys | Observers walk along predetermined paths (transects) and record all birds encountered within a specified distance. | Systematic coverage of habitats, suitable for diverse ecosystems. | [4,13,17,18] |
| Drone technology | Unmanned aerial vehicles equipped with cameras or sensors conduct aerial surveys for bird counting. | Efficient for large-scale surveys, accessing difficult terrains. | [6,8,19] |
| Point counts | Observers station themselves at predetermined points and record all birds seen or heard within a specified time. | Simple, cost-effective, and provides data on bird abundance and distribution. | [13,17,18] |
| Automated acoustic monitoring | Autonomous recording units or smartphone apps capture bird vocalizations, and automated software analyzes the recordings to identify and count the species. | Continuous monitoring, especially useful for nocturnal species. | [20,21] |
| Remote sensing and satellite imagery | High-resolution satellite imagery or aerial photography is analyzed to identify and estimate bird populations based on habitat characteristics. | Large-scale monitoring, useful for waterfowl and colonial nesting species. | [22,23] |

Point counting is one important method to monitor populations of birds from time to time. It is usually performed by taking several pictures or videos of a flock of birds from an observatory station for about 3–10 min, depending on the purpose of the study [24,25]. Although important, the process is tedious, consisting of manually counting the birds one by one. Moreover, manual counting is subject to error and is time-consuming, as there are several important factors, such as individual knowledge, high bird numbers, and many image samples [26]. Due to the recent breakthroughs in AI-based computer vision, it is possible to utilize AI to assist bird observation and data analysis. However, there are several limitations in its use, especially the limitations of the researchers' knowledge about operating the necessary software to prepare the dataset and the training of the AI neural network, as well as the need for high-end computers to efficiently train/process these images [27,28]. Additionally, depending on the algorithm, there are some AIs that are not

suitable for detecting certain objects, resulting in recognition errors, which is why some researchers prefer to observe them manually, particularly in specific cases in which it is hard to distinguish between the objects of interest and the background.

ImageJ is an open-source platform used for image processing and software analysis that was developed by the National Institutes of Health (NIH) and that is extensively used in various fields, including biology, medicine, and material science [29]. ImageJ contains numerous plugins that are useful for image analysis, including filtering and normalization, thresholding, object identification, and particle analysis [30–32]. It is also equipped with batch processing, which helps with analyzing multiple images simultaneously. The usage of ImageJ for bird counting has been proposed previously by Hurford [33] and Spoorthy et al. [26]. They demonstrated the ability of the Particle Analyzer function in ImageJ to count the number of birds from several images after applying a thresholding method. Valle et al. also reported the analysis of greater flamingo flocks using the Find Maxima method in ImageJ from drone images [34]. Although these studies highlighted the potency of both the Particle Analyzer and Find Maxima functions in ImageJ, these methods have several limitations, such as a color contrast difference between birds and backgrounds and the possibility of overcounting/undercounting due to the bird size. ImageJ is also well known for having several image segmentation tools and plugins. The most common is the Watershed algorithm, which is commonly used in cell studies to separate touching or overlapping objects in binary images [35]. For example, some studies have used Watershed segmentation to separate overlapping blood cells [36] and for the observation of neurodegeneration in *Drosophila* [37]. Trainable WEKA (Waikato Environment for Knowledge Analysis) segmentation is a plugin available in ImageJ. This plugin uses a combination of machine learning algorithms and a set of image features to produce pixel-based segmentations [38]. Previously, Lormand et al. used WEKA segmentation to observe the crystal size distribution in volcanic rocks [39], while Salum et al. used it to determine the droplet size in an emulsion [40], and it has also been used to classify and count the numbers of plants [41] and cells [38]. TrackMate is an ImageJ plugin developed by Tinevez et al. for particle tracking based on the particle pixel size [42]. Although it was built for particle tracking, in application, it has also been used for tracking blood cells [43,44], lysosomes [45,46], and even *Drosophila* movement [47,48].

Thus, based on prior studies, this study proposed the use of TrackMate, Watershed, and trainable WEKA segmentation as alternative semi-automatic methods for counting the numbers of birds and pinpointing their positions. The capabilities of these methods were then compared to those of previously tested methods: the Particle Analyzer, Find Maxima, and true-value (manual counting) methods. The bird count data were obtained from videos, while several image frames were sampled from the videos and were used for manual pinpointing to compare the coordinates obtained from all the methods to test the sensitivity and precision.

## 2. Materials and Methods

### 2.1. Sample Video Datasets and Preprocessing

In this study, several videos were used to represent case-to-case studies in recording the movements of birds, and they mainly came from online platforms. The first case represents the bird movement in unidirectional form. The first video for the first case was obtained from Kang's nature channel on YouTube (available online: https://www.youtube.com/watch?v=5OaklG1qLSU, accessed on 10 January 2024). The video was cut from frame 415 to 470. The second video was obtained from the same YouTube channel (available online: https://www.youtube.com/watch?v=Plf1A6qeGcQ, accessed on 10 January 2024) and was cut from frame 1496 to 1573. The second case represents a case with a low background-to-object contrast ratio, and the video was obtained from Peter Chen's YouTube channel (available online: https://www.youtube.com/watch?v=t7xPuQVGjuY, accessed on 26 December 2023) and was cut from frame 445 to 544. The third case represents the multidirectional movement of birds in high numbers with a respectable background-to-

object contrast ratio, and the videos were obtained from Suuuhus' channel (available online: https://www.youtube.com/watch?v=PjbX5x9ZB8w, accessed on 26 December 2023), cut from frame 0 to 200, and from the Free High-Quality Documentaries' channel YouTube source (available online: https://www.youtube.com/watch?v=uGrvvVqsdrk, accessed on 26 December 2023), cut from frame 31,800 to 31,999. The last case represents a very high number of birds (>1000). The video for this case was obtained from Claireonline's channel (available online: https://www.youtube.com/watch?v=m6YDhVeW5Kc, accessed on 26 December 2023) and it was cut from frame 101 until 260. All videos were cut using VirtualDub2 build 44282 software and were saved in ImageJ-supported format (available online: https://sourceforge.net/projects/vdfiltermod/, accessed on 26 December 2023).

Ten frames from each video were selected randomly, the birds' locations were pinpointed, and their numbers were counted with the assistance of the Cell Counter tool in the FIJI distribution of ImageJ version 1.54f (https://imagej.net/software/fiji/, accessed on 22 August 2023) [42]. This method is deemed as the true value, as it is based on manual observation. Also using ImageJ, the videos were also preprocessed before detection using the counting methods. Firstly, the videos were converted into grayscale (Image > Type > 8-Bit). In case there were a lot of background objects, we used the "Convert to Mask" tool to remove most of the background (Process > Binary > Convert to Mask) with the intermodes filter. All processes were conducted on a computer with i7-9700K CPU with a GTX 1060 6 GB graphics card and 64 GB of RAM.

### 2.2. TrackMate Method

TrackMate plugin version 7.11.1 was used to detect and count the bird numbers from individual frames. After preprocessing, the TrackMate plugin was run (Plugins > Tracking > TrackMate), and the Difference of Gaussian (DoG) Detector was used to detect the birds, which is a faster detector compared to the Laplacian of Gaussian (LoG) plugin available in TrackMate [49]. For the analysis, the quality threshold was set to 0.5 based on previous pre-testing, while the object diameter was set according to the videos. Finally, the detected spots at each frame were counted as birds, while the XY coordinates were also exported for further endpoint calculations.

### 2.3. Particle Analyzer Method

The Particle Analyzer method mainly uses the available Analyze Particle tool (Analyze > Analyze Particles...) available in ImageJ. This tool is used to count and measure objects based on their size and circularity. The threshold was preemptively applied to the image to mark the objects from the background, and the birds were measured/counted using the Analyze Particle tool by setting a size threshold (in pixels). The number of birds and their XY coordinates were then exported as endpoints.

### 2.4. Watershed Method

Watershed is a technique used to separate overlapped or connected objects from each other. This technique is commonly used to separate connected cells from staining results. Watershed is available as a tool in ImageJ (Process > Binary > Watershed). To apply Watershed, firstly, the image is converted to a binary, dark object (intensity value = 0) on a white background (intensity value = 255), and it is only then that Watershed can be applied. The Analyze Particle tool was used to count the number of birds and export their XY coordinates.

### 2.5. Find Maxima Method

The Find Maxima tool (Process > Find Maxima...) detects the spots with intensities above a set threshold. In order to increase the detection accuracy, the image was converted into a binary image earlier in the process. For the tested images, the prominence was set to >10, and the results were exported as the number of birds and their XY coordinates.

### 2.6. Trainable WEKA Segmentation Method

Trainable WEKA segmentation is an ImageJ plugin that uses machine learning to produce pixel-based segmentations. To use WEKA, it is imperative to train a model suitable for the image/video that needs to be segmented. In this study, all the videos had their backgrounds subtracted and converted to binary images, and 10 images from each video were used to create the training dataset. The training was conducted to segment the objects from the background from the binary images. Gaussian blur, Sobel filter, Hessian, Difference of Gaussian, and Membrane projections were used as the training features to train the model. The trained model was then exported and used to segment the videos. Finally, similar to the previous mentioned methods, the Analyze Particle tool was used to obtain the number of birds and export their XY coordinates.

### 2.7. Sensitivity, Precision, and F1 Score Calculation Using Python

Homebrewed Python scripts were used to determine the sensitivity and precision of each method by comparing the detected centroid XY coordinates to the true value. The first Python script was used to overlay the coordinates obtained from the tested methods with the true value (Supplementary File S1), while the second Python script was used to calculate the sensitivity and precision (Supplementary File S2). The second script was iterated through the Excel file containing the coordinates from the true value and tested methods, comparing all of them within a set "distance threshold", as true-value manual pinpointing might not be accurate for the object centroid. Afterward, one coordinate located within the "distance threshold" from the true value was deemed as the true positive, while the other coordinates located within the "distance threshold" and the other detected objects outside of the distance threshold were grouped as false positives. Finally, if the methods did not detect any centroid within the "distance threshold", it was grouped as a false negative for the respective method. We tested the sensitivity (also known as the recall), precision, and F1 score of all the tested methods on 10 images using the following equations:

$$\text{Sensitivity} = \frac{\text{True positives}}{\text{True positives} + \text{False negatives}}$$

$$\text{Precision} = \frac{\text{True positives}}{\text{True positives} + \text{False positives}}$$

Afterward, the F1 score was calculated to check the overall reliability of each method:

$$\text{F1 Score} = \frac{2(\text{Precision} * \text{Sensitivity})}{(\text{Precision} + \text{Sensitivity})}$$

These metrics are commonly used to evaluate the performances of models. In this case, the tested methods were compared to manual counting. Sensitivity was used to measure the ability of the tested method to identify the right object, while precision measured the accuracy of the method to differentiate between the object of interest and the background noise or other objects. Finally, the F1 score is the harmonic mean between the sensitivity and precision, as it integrates the values obtained from the sensitivity and precision into a single value [50].

### 2.8. Statistical Calculation

Statistical analysis was performed using GraphPad Prism 8 (Graphpad Holdings, LCC, San Diego, CA, USA). The result of each method was compared to the true value using Deming linear regression. Deming linear regression is a commonly used statistical test for comparing a method to a preexisting method, and it is preferable to ordinary linear regression (OLR) because it overcomes the bias problem present in OLR. In the Deming linear regression analysis, the X value was the true value obtained from manual counting, while the Y values were the values obtained from the tested methods, which were compared

to the manual counting value. The ideal result for Deming linear regression is slope = 1, with bias = 0 [51].

## 3. Results

### 3.1. Case 1: Birds Moving in Unidirectional Movement

In the first case, we used a video of cattle egrets moving unidirectionally with the sky and sea in the background (Figure 1A and Video S1). Figure 1B shows a representation of the birds' positions obtained from the first frame of the video from the five different testing methods. From the image, it seems most of the pinpointed coordinates are closely related to each other; therefore, the distance threshold for the sensitivity and precision calculation was set to 15 pixels. Both the frame-by-frame bird-counting and Deming regression data showed that all five tested methods showed similar counting results for the cattle egrets (Figure 1C,D). The Deming regression graph showed that TrackMate is the most suitable, as it is located close to the identity line (highlighted with the blue line). However, the equation result showed that Watershed had the closest slope value to 1 ($Y = 1.013x - 2.955$, Table 2). However, by also considering the results of the sensitivity and precision test, TrackMate seems to be the preferable detection method, as it had higher sensitivity, precision, and F1 score values compared to Watershed, which showed the lowest sensitivity ($0.954 \pm 0.036$), precision ($0.962 \pm 0.032$), and F1 score ($0.957 \pm 0.023$) values. Additionally, the Find Maxima method can also be used as an alternative counting method in this case, with respectable detection/counting results, followed by Particle Analyzer and WEKA, which have identical values (Table 2).

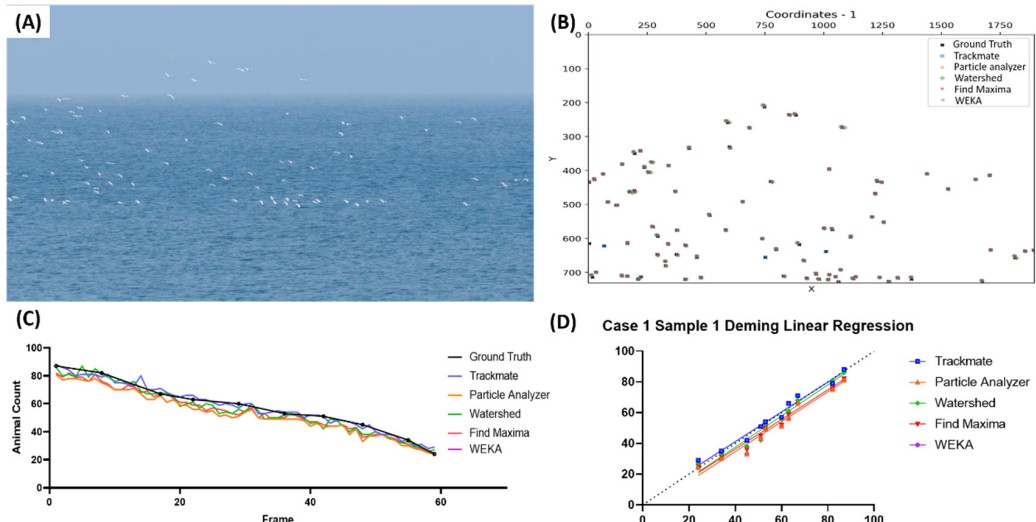

**Figure 1.** Testing five different ImageJ-based methods for cattle egret number counting from video dataset. Reference image of the first video in the first case (**A**) and the coordination plot showing the positions of the pinpointed coordinates in the first frame of the video from multiple methods and the true value (**B**). Reported bird number from frame-by-frame observation between the true value and every tested method (dots show the positions of the frames compared in the test endpoints) (**C**), and Deming linear regression results of multiple tested methods compared to true value (**D**).

**Table 2.** Statistical test result for cattle egret number counting.

| Method | Deming Regression | Sensitivity * | Precision * | F1 Score * |
|---|---|---|---|---|
| TrackMate | $Y = 0.967x + 2.460$ | $0.954 \pm 0.036$ | $0.962 \pm 0.032$ | $0.957 \pm 0.023$ |
| Particle Analyzer | $Y = 0.971x - 4.166$ | $0.905 \pm 0.038$ | $0.990 \pm 0.025$ | $0.945 \pm 0.028$ |
| Watershed | $Y = 1.013x - 2.955$ | $0.870 \pm 0.017$ | $0.938 \pm 0.047$ | $0.902 \pm 0.027$ |
| Find Maxima | $Y = 0.962x - 2.167$ | $0.931 \pm 0.031$ | $0.985 \pm 0.023$ | $0.957 \pm 0.023$ |
| WEKA | $Y = 0.971x - 4.166$ | $0.905 \pm 0.038$ | $0.990 \pm 0.025$ | $0.945 \pm 0.028$ |

* Data are presented as mean $\pm$ SD.

The next example showed geese moving unidirectionally in Estonia with a background separated into three parts, namely, the sky, faraway trees, and ocean waves (Figure 2A and Video S2). We extracted the detected coordinates from each bird in Figure 2B. In order to calculate the sensitivity and precision, the distance threshold was set to 30 pixels, as some true-value selections were positioned further from the centroids. It was also observed that the number of birds mostly stayed the same throughout the video. However, the Watershed method seemed to overcount the number of birds by around 1.5 times, which was further shown in the Deming linear regression result (Figure 2C,D, highlighted with the green color). By observing the Deming regression analysis results, none of the methods showed a slope value close to 1, with TrackMate being the closest one (Y = 2.505x − 55.71, Table 3). The sensitivity (0.897 ± 0.030), precision (0.988 ± 0.015), and F1 score (0.940 ± 0.022) values also support TrackMate as the best method in this tested case, as it detected nearly 90% of the birds while showing minimal false-positive detection. Meanwhile, the high sensitivity value (0.967 ± 0.036) of the Watershed method was not supported by its low precision (0.654 ± 0.035) due to its high false-positive number (Table 3).

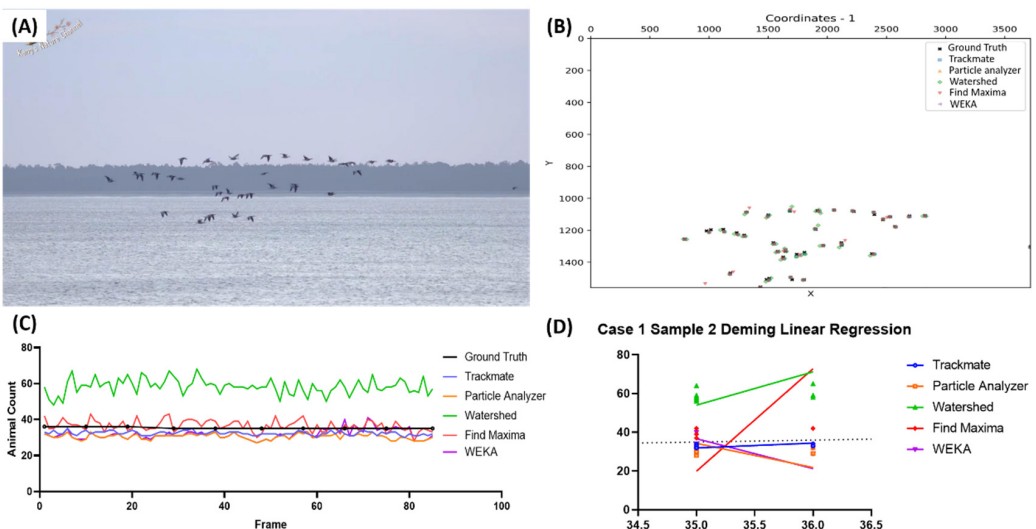

**Figure 2.** Testing five different ImageJ-based methods for goose number counting from video dataset. Reference image of the first video in the first case (**A**) and the coordination plot showing the positions of the pinpointed coordinates in the first frame of the video from multiple methods and true value (**B**). Reported bird numbers from frame-by-frame observation between the true value and every tested method (dots show the positions of the frames compared in the test endpoints) (**C**), and Deming linear regression results of multiple tested methods compared to true value (**D**).

**Table 3.** Statistical test result for goose number counting.

| Method | Deming Regression | Sensitivity * | Precision * | F1 Score * |
|---|---|---|---|---|
| TrackMate | Y = 2.505x − 55.71 | 0.897 ± 0.030 | 0.988 ± 0.015 | 0.940 ± 0.022 |
| Particle Analyzer | Y = −12.47x + 470.8 | 0.823 ± 0.077 | 0.967 ± 0.022 | 0.888 ± 0.054 |
| Watershed | Y = 17.22x − 548.6 | 0.967 ± 0.036 | 0.654 ± 0.035 | 0.780 ± 0.025 |
| Find Maxima | Y = 52.90x − 1832 | 0.761 ± 0.088 | 0.837 ± 0.061 | 0.794 ± 0.063 |
| WEKA | Y = −15.43x + 576.8 | 0.823 ± 0.077 | 0.967 ± 0.022 | 0.888 ± 0.054 |

* Data are presented as mean ± SD.

### 3.2. Case 2: Birds Moving in Unidirectional Direction with Filled Background

The second case tested the performances of the proposed and previous methods in the case in which the background is filled with objects (Figure 3A). In Video S3, cattle egrets are flying over a complex background filled with green-colored trees. Even though the trees are visually different, most image detection workflows in ImageJ always involve conversion to an eight-bit grayscale; therefore, there will be some lost features during the

process. To salvage this, we resorted to converting the videos to mask, using the "Convert to Mask" tool, setting the filter to intermodes before detection. The results of the detected bird coordinates are shown in Figure 3B.

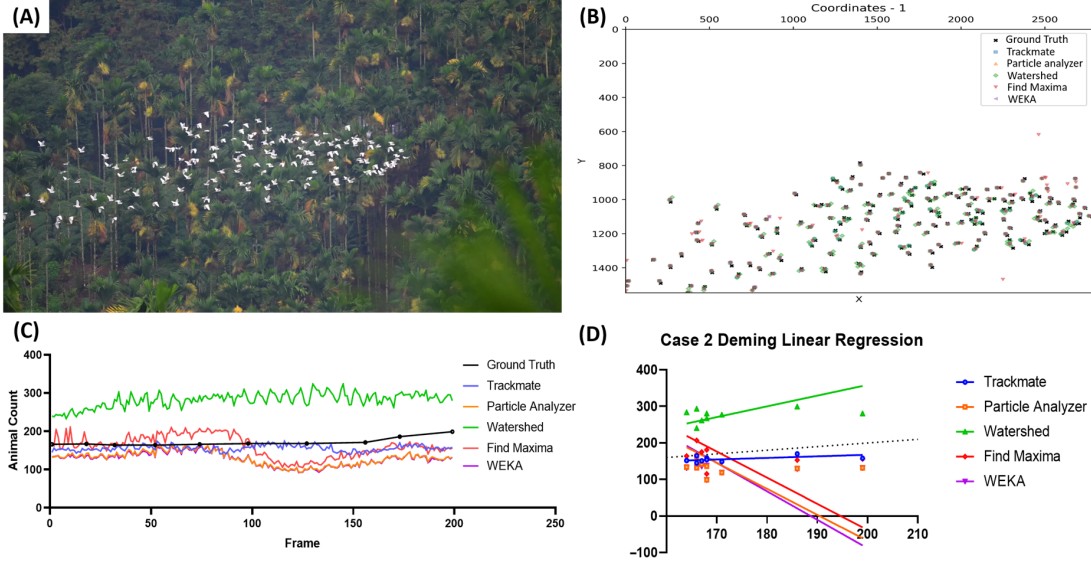

**Figure 3.** Testing five different ImageJ-based methods for cattle egret number counting from video dataset. Reference frame of the sample video used in the second case test (**A**) and the coordination plot showing the positions of the pinpointed coordinates in the first frame of the video from multiple methods and true value (**B**). Reported bird numbers from frame-by-frame observation between the true value and every tested method (dots show the positions of the frames compared in the test endpoints) (**C**), and Deming linear regression results of multiple tested methods compared to true value (**D**).

The distance threshold was set to 30 to take into account the true-value pinpointed coordinates. From the image, we can observe that most of the methods have comparable detection results to the true value, excluding the Find Maxima method. This result highlights the limitation of the ImageJ Find Maxima method, as it lacks the option to set the minimum/maximum size. It is possible to classify background objects as objects of interest. However, the frame-by-frame counting results showed that Watershed overcounted the number of birds, similar to the result in the first case (Figure 3C, highlighted with the green color). In the Deming regression results, TrackMate undercounted the counting results compared to the true value (highlighted with the blue color), while the Watershed showed overcounting (highlighted with the green color) and the other methods showed undercounting for high bird numbers, most likely due to overlapping (Figure 3D and Table 4). The results of the precision, sensitivity, and F1 score showed that none of the methods that we tested showed acceptable values, with TrackMate, Particle Analyzer, and WEKA having high precision ($0.863 \pm 0.141$, $0.851 \pm 0.113$, and $0.832 \pm 0.117$, respectively) but low sensitivity ($0.683 \pm 0.153$, $0.594 \pm 0.122$, and $0.581 \pm 0.119$, respectively) and Watershed having high sensitivity ($0.895 \pm 0.098$) but low precision ($0.597 \pm 0.062$) (Table 4).

**Table 4.** Statistical test result for cattle egret number counting.

| Method | Deming Regression | Sensitivity * | Precision * | F1 Score * |
|---|---|---|---|---|
| TrackMate | Y = 0.4325x + 81.15 | $0.683 \pm 0.153$ | $0.863 \pm 0.141$ | $0.761 \pm 0.149$ |
| Particle Analyzer | Y = −7.115x + 1355 | $0.594 \pm 0.122$ | $0.851 \pm 0.113$ | $0.697 \pm 0.121$ |
| Watershed | Y = 2.938x − 228.8 | $0.895 \pm 0.098$ | $0.597 \pm 0.062$ | $0.716 \pm 0.071$ |
| Find Maxima | Y = −7.160x + 1394 | $0.604 \pm 0.147$ | $0.715 \pm 0.090$ | $0.643 \pm 0.087$ |
| WEKA | Y = −7.793x + 1470 | $0.581 \pm 0.119$ | $0.832 \pm 0.117$ | $0.681 \pm 0.118$ |

* Data are presented as mean ± SD.

### 3.3. Case 3: Multidirectional Movement of Birds in High Numbers

The third case represents the condition in which Chinese sparrowhawks (*Accipiter soloensis*) moved in a multidirectional manner (Video S4). A sample image of the tested video is presented in Figure 4A. The distance threshold for the sensitivity and precision calculation was set to 10 pixels. From Figure 4B, most of the tested methods showed high accuracy compared to the true value. However, the result of a further investigation of the frame-by-frame counting did not support the previous observation (Figure 4C). The Watershed method was found to overcount the bird-counting results in this sample video (highlighted with the green color), while WEKA slightly undercounted the counting results (highlighted with the pink color). Further observation of the Deming regression equation also supports the previous result (Figure 4D). From all the tested methods, TrackMate shows the most preferable result, as its slope is close to 1 (Y = 1.029x − 6.137) and it is also supported with high sensitivity (0.981 ± 0.010), precision (0.995 ± 0.006), and F1 score (0.988 ± 0.008) values (Table 5). In addition, the Particle Analyzer and Find Maxima also showed decent results, making them alternatives for counting birds in this type of video.

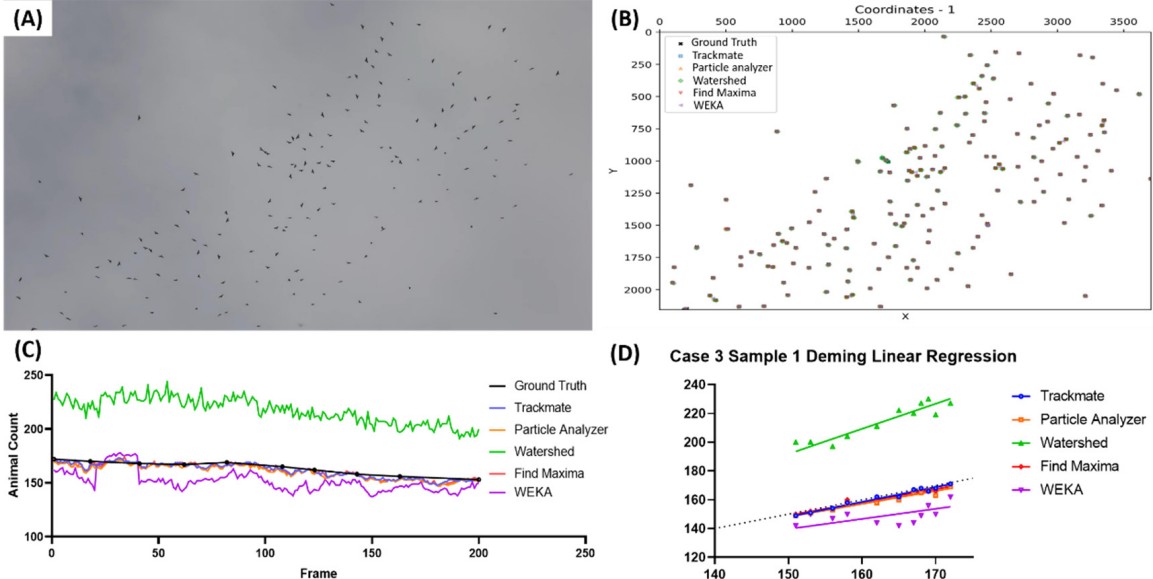

**Figure 4.** Testing five different ImageJ-based methods for Chinese sparrowhawk number counting from video dataset. Reference frame of the first sample video used in the third case (**A**) and the coordination plot showing the positions of the pinpointed coordinates in the first frame of the video from multiple methods and true value (**B**). Reported bird numbers from frame-by-frame observation between the true value and every tested method (dots show the positions of the frames compared in the test endpoints) (**C**), and Deming linear regression results of multiple tested methods compared to true value (**D**).

**Table 5.** Statistical test results for Chinese sparrowhawk number counting.

| Method | Deming Regression | Sensitivity * | Precision * | F1 Score * |
|---|---|---|---|---|
| TrackMate | Y = 1.029x − 6.137 | 0.981 ± 0.010 | 0.995 ± 0.006 | 0.988 ± 0.008 |
| Particle Analyzer | Y = 0.9027x + 12.75 | 0.972 ± 0.011 | 0.992 ± 0.007 | 0.982 ± 0.07 |
| Watershed | Y = 1.747x − 70.09 | 0.990 ± 0.009 | 0.728 ± 0.023 | 0.839 ± 0.015 |
| Find Maxima | Y = 0.9163x + 11.71 | 0.971 ± 0.011 | 0.991 ± 0.006 | 0.981 ± 0.007 |
| WEKA | Y = 0.7037x + 34.16 | 0.747 ± 0.061 | 0.920 ± 0.030 | 0.824 ± 0.048 |

* Data are presented as mean ± SD.

The second sample video representation is shown in Figure 5A. The video showed birds from the *Ciconiidae* family flying in a circular motion due to a warm air current (Video S5). The representation of the birds' pinpointed coordinates is presented in Figure 5B. It

was observed that there were extra Watershed coordinates compared to the other methods. The distance threshold for the sensitivity and precision calculation was set to 25 for this video. Watershed overcounting can also be observed in Figure 5C (highlighted with the green color). The rest of the tested methods showed undercounting of the bird count. From the Deming regression (Figure 5D) equation, Watershed showed the most similar result to the true value (Y = 0.9454x + 37.18), but the result is not supported by its relatively low precision (0.812 ± 0.038), while the other methods tended to undercount the bird-counting result, as mentioned previously (Table 6). In this case, TrackMate might also be a preferable method, as its sensitivity (0.846 ± 0.059) and precision (0.959 ± 0.019) values are quite balanced, even though Particle Analyzer and WEKA showed higher F1 scores.

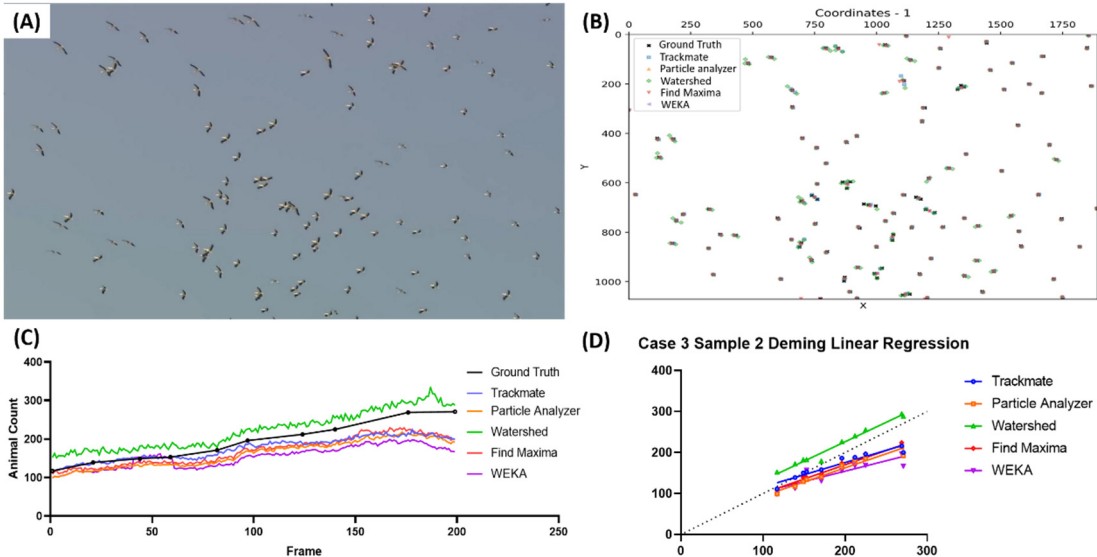

**Figure 5.** Testing five different ImageJ-based methods for *Ciconiidae* number counting from video dataset. Reference frame of the second sample video used in the third case (**A**) and the coordination plot showing the positions of the pinpointed coordinates in the first frame of the video from multiple methods and true value (**B**). Reported bird numbers from frame-by-frame observation between the true value and every tested method (dots show the positions of the frames compared in the test endpoints) (**C**), and Deming linear regression results of multiple tested methods compared to true value (**D**).

**Table 6.** Statistical test results for *Ciconiidae* number counting.

| Method | Deming Regression | Sensitivity * | Precision * | F1 Score * |
|---|---|---|---|---|
| TrackMate | Y = 0.5946x + 56.60 | 0.846 ± 0.059 | 0.959 ± 0.019 | 0.897 ± 0.032 |
| Particle Analyzer | Y = 0.6858x + 25.46 | 0.823 ± 0.049 | 0.994 ± 0.004 | 0.900 ± 0.031 |
| Watershed | Y = 0.9454x + 37.18 | 0.892 ± 0.039 | 0.812 ± 0.038 | 0.849 ± 0.024 |
| Find Maxima | Y = 0.6944x + 31.23 | 0.808 ± 0.047 | 0.963 ± 0.021 | 0.878 ± 0.033 |
| WEKA | Y = 0.5058x + 53.39 | 0.823 ± 0.047 | 0.994 ± 0.005 | 0.900 ± 0.030 |

\* Data are presented as mean ± SD.

### 3.4. Case 4: Very High Number of Birds (>1000)

The last case highlighted the abilities of the proposed methods to detect birds in very high numbers (>1000), specifically birds from the *Sturnidae* family during murmuration (Video S6). The image representation of the video used is presented in Figure 6A, and the image shows a very high number of birds in contrast to the background. The representations of the bird positions from all the tested methods are decently accurate, as seen in Figure 6B. However, due to the high number of birds in the image, it is hard to accurately pinpoint the detection quality from this image alone. The frame-by-frame counting results (Figure 6C) showed that, in the early frames, most of the methods showed comparable

counting results to the true value. However, in the later frames, it seems that all the methods undercounted the bird count, with Watershed (highlighted with the green color) showing the closest counting result to the true value from all the proposed methods. As expected, Watershed showed the best Deming regression equation compared to the other methods (Y = 0.8638x + 260.3), followed by Particle Analyzer (Y = 0.8270x + 216.0), Find Maxima (Y = 0.8270x + 216.0), TrackMate (Y = 0.5823x + 488.7), and lastly, WEKA (Y = 0.3557x + 822.8) (Figure 6D). The sensitivity, precision, and F1 score results also support the Deming regression results accordingly (Table 7).

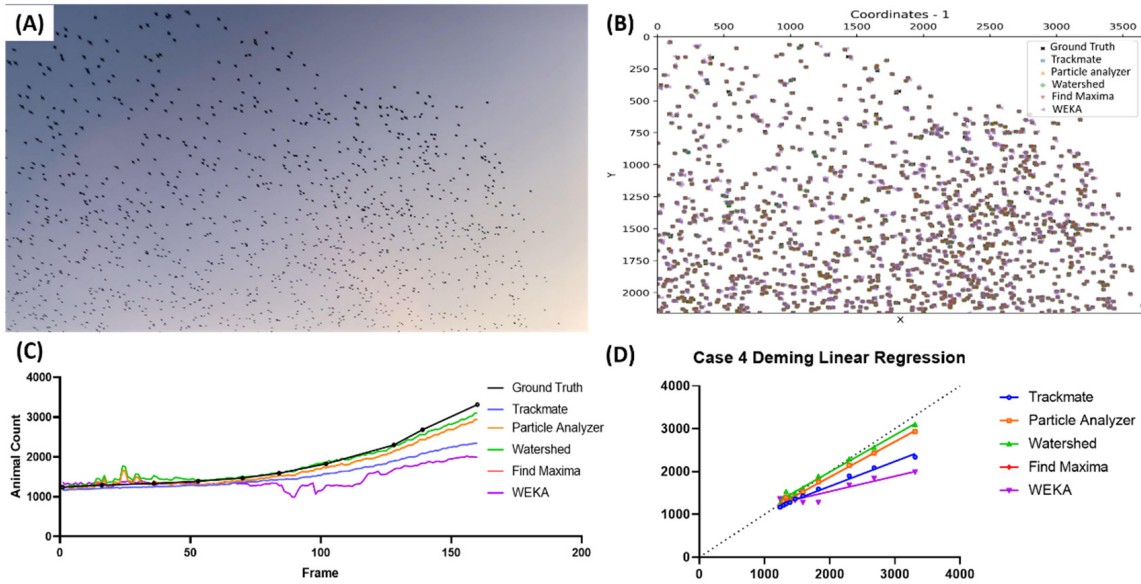

**Figure 6.** Testing five different ImageJ-based methods for *Sturnidae* number counting from video dataset. Reference frame of the sample video used in the fourth case (**A**) and the coordination plot showing the positions of the pinpointed coordinates in the first frame of the video from multiple methods and true value (**B**). Reported bird numbers from frame-by-frame observation between the true value and every tested method (dots show the positions of the frames compared in the test endpoints) (**C**), and Deming linear regression results of multiple tested methods compared to true value (**D**).

**Table 7.** Statistical test result for *Sturnidae* number counting.

| Method | Deming Regression | Sensitivity * | Precision * | F1 Score * |
|---|---|---|---|---|
| TrackMate | Y = 0.5823x + 488.7 | 0.858 ± 0.067 | 0.997 ± 0.003 | 0.921 ± 0.040 |
| Particle Analyzer | Y = 0.8270x + 216.0 | 0.911 ± 0.029 | 0.983 ± 0.007 | 0.945 ± 0.018 |
| Watershed | Y = 0.8638x + 260.3 | 0.938 ± 0.028 | 0.973 ± 0.010 | 0.955 ± 0.016 |
| Find Maxima | Y = 0.8270x + 216.0 | 0.910 ± 0.030 | 0.983 ± 0.007 | 0.945 ± 0.018 |
| WEKA | Y = 0.3557x + 822.8 | 0.552 ± 0.045 | 0.653 ± 0.080 | 0.593 ± 0.017 |

* Data are presented as mean ± SD.

## 4. Discussion

From the four major cases tested in this study, we found that every tested method for bird counting from the video dataset has its advantages and limitations. In Video S1, used in the first case (cattle egret migrating unidirectionally above the sea), we found that all the methods showed comparable counting and detection results to the true value. However, the Watershed method showed overcounting in most cases. Previously, Watershed has been proposed as a common method used for separation, especially in cell-related studies [36,52]. However, in this study, the birds were normally not circular-/oval-shaped; thus, this might have compromised the object separation of the Watershed method. This result is supported by the Watershed's low precision in most of the tests conducted in this study, except the

fourth case, in which the birds were abundant, which might have led to overlapping and the non-prominence of their features. Additionally, the abundance of birds might have also played a role in improving the precision value. Thus, we propose that the Watershed segmentation method might assist in separating highly overlapping birds to increase its sensitivity and precision.

We found that the Maxima method performed well in most of the tested videos, showing high sensitivity and precision values in most of the tested cases, except in Video S2 for the goose migration, in which it showed low precision and sensitivity values with a very high Deming regression slope. The marine background of this video seemed to interfere with the Find Maxima results, as the precision was compromised further due to the existing background artifacts even after the background removal.

Particle Analyzer and WEKA had several identical results in our tests. The similarity might have happened due to the use of ImageJ's Particle Analyzer tool to count the numbers of birds and pinpoint their coordinates. In the other cases for which the data are not identical, Particle Analyzer generally showed higher precision and sensitivity. Even though WEKA is a segmentation tool, it did not seem to segment the birds and it created overcounting, similar to the Watershed method. Based on our observation, WEKA segmentation seems to overestimate the size of detected objects. Therefore, it might create a pseudo-connection between objects, possibly creating false-negative results. The inability to separate overlapping objects is a common problem when using the Particle Analyzer and Find Maxima methods, as there is no built-in segmentation tool in their workflows [33]. WEKA seems to have the same problem, which was reported in a previous study [40].

TrackMate was initially designed for particle tracking in ImageJ. In this study, we did not fully utilize and assess the tracking capability of TrackMate. We only tested its accuracy in recognizing individual objects, pinpointing their coordinates, and possibly recognizing the connected/overlapped objects that separate them. TrackMate showed a good performance in most of the conducted tests, with high precision and relatively high sensitivity values. These results suggest that TrackMate did not often recognize the background as the object of interest. However, there is a limitation in recognizing the object of interest. TrackMate's poor performance in the fourth case (Sturnidae counting) might have been due to the smaller size of the birds in the later frames of the video, creating an identity loss, thereby making them unrecognizable to the system (Figure 7). However, this case can probably be fixed by splitting the video to set a different size threshold for each part.

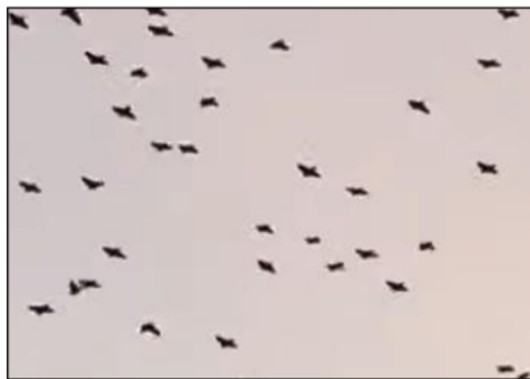 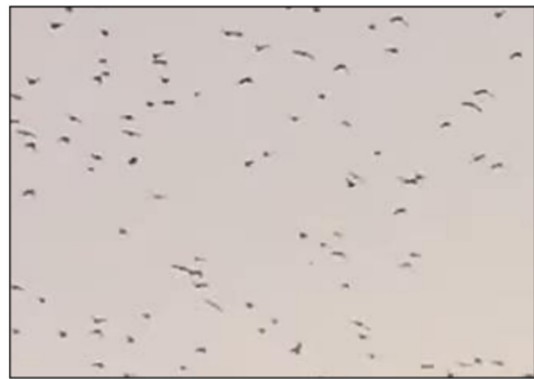

**Figure 7.** Comparison of bird sizes taken from the fourth case (*Sturnidae*) at the same position during the first (**left**) and last (**right**) frames of the video.

Lastly, we would like to highlight the cattle egret migration with the complex background (Video S3). In this case, none of the tested methods showed acceptable results with relatively low sensitivity and high precision. This result means that they failed to detect most of the birds, or they had a lot of false negatives. However, Watershed had high sensitivity and low precision, similar to its results in the other cases. Thus, none of the methods we tested are recommended for use for this particular case. To overcome this limitation,

a deep learning-based method is proposed to obtain better bird-number-counting results with complex backgrounds in the future.

## 5. Conclusions

By taking into account the counting results and Deming linear regression results, we found that most of the tested counting methods showed different counting accuracies depending on the object-to-background contrast ratio, presence of background objects, number of birds, and coincidence of overlapping birds. The most important finding for this study is that the TrackMate method has been validated as the preferred method for counting the numbers of birds and pinpointing their coordinates in most cases. Additionally, TrackMate also has the unexplored ability to track their movements, which might be interesting for further study. Particle Analyzer and Find Maxima are also good alternatives to TrackMate, as they come preinstalled in ImageJ, with respectable results, depending on the background. WEKA is not recommended due to the fact that it overestimated the object size in our tests, creating a pseudo-connection between the objects and thereby undercounting the final results. Lastly, Watershed segmentation must be used with caution for bird counting, as it is not fully suitable for segmenting birds due to the shape differences among and physical characteristics of the birds, especially the colors of their bodies. In conclusion, we tested the bird-counting performances of five different counting/detection methods on the ImageJ platform for the first time. Among the five tested methods, TrackMate was validated as the most suitable method for bird counting, with superior sensitivity, precision, and F1 score values (Table 8).

**Table 8.** The advantages and disadvantages of each tested method for bird counting.

| Methods | TrackMate | Particle Analyzer | Find Maxima | Watershed | Trainable WEKA Segmentation |
|---|---|---|---|---|---|
| Advantages | Can be used to count birds with high precision and respectable sensitivity. | Size thresholding helps differentiate birds from unwanted background objects. | The fastest and simplest method to count birds. | Segmentation helps to count connecting/overlapping small birds. | Able to output similar results to Particle Analyzer. |
| Disadvantages | Limited capability in detecting objects with dynamic sizes in the video; takes some time to process the video (depends on the video resolution and duration). | No segmentation; cannot separate connecting/overlapping objects; might need some experience and adjustments in setting the size threshold to obtain the best results. | Unable to measure the object size; no thresholding to differentiate birds from unwanted objects. | The segmentation algorithm is mostly not suitable for birds, most of the time resulting in the overcounting of the bird number (low precision). | Difficulty in detecting objects with significant size differences, and there were times when the segmentation actually increased the size of the objects, creating a pseudo-connection, reducing the bird count; it takes time to train the detection model. |

**Supplementary Materials:** The following supporting information can be downloaded at https://www.mdpi.com/article/10.3390/inventions9030055/s1. Supplementary File S1. Coordinates overlay python script, Supplementary File S2. Parameter calculations python script, Supplementary Video S1. Case 1 Video 1, Supplementary Video S2. Case 1 Video 2, Supplementary Video S3. Case 2, Supplementary Video S4, Case 3 Video 1, Supplementary Video S5, Case 3 Video 2, Supplementary Video S6. Case 4.

**Author Contributions:** K.A.K.: writing—original draft; formal analysis; methodology; data curation; visualization; investigation; validation. F.S.: writing—original draft; formal analysis; methodology;

data curation; visualization. C.T.L.: methodology; software; validation. M.J.M.R.: writing—review and editing. T.-S.C.: conceptualization; supervision; project administration. C.-D.H.: conceptualization; writing—original draft; writing—review and editing; investigation; supervision; resources; project administration. All authors have read and agreed to the published version of the manuscript.

**Funding:** The authors received no financial support for the research, authorship, or publication of this article.

**Data Availability Statement:** Supplementary data can be found in the Google drive https://drive.google.com/drive/folders/1OFmHlY0lofqYJX39q06hjpRz2avEmaTf?usp=sharing (accessed on 2 May 2024).

**Acknowledgments:** We would like to thank Cao Thang Luong for developing the Python script for calculating the endpoints. We appreciate Marri Jmelou Roldan from the University of Santo Tomas for providing English editing to enhance the quality of the paper.

**Conflicts of Interest:** The authors declare no conflicts of interest.

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
