# Peer review of "Performance Comparison of Five Methods Available in ImageJ for Bird Counting and Detection from Video Datasets"

_inventions, doi:10.3390/inventions9030055_

Round 1
Reviewer 1 Report
Comments and Suggestions for Authors
Dear Authors,
1. The MS compares five plugins of ImageJ, a software developed for the analysis of micro- to macro-objects. Consequently, the MS is a long review of the capacities of the five plugins, and in scientific terms, it has a low level of novelty.
2. This kind of review can help ornithologists choose among the many similar applications. To achieve this, it would be useful to use some pointing system or list the pros and cons of each plugin. In the text, the reader can have difficulties following their presentations and understanding the advantages or disadvantages.
3. The introduction of the MS is lengthy, and the importance of birds, and general conservation topics are far beyond the scope of this text. A concise introduction focusing on bird population assessments is right enough.
4. The authors use counting, estimation, and monitoring as synonims. In the wildlife conservation (and related statistical) literature, these words have definitions, and are carefully used. This MS is a text about bird counting with the assumption that, on digital pictures, birds can be much better enumerated than with other kinds of methods. Although the theoretical assumption is 100% countability/visibility, the final results show that, depending on the plugin (for the user, each plugin is a blackbox), over- or underestimation is typical. Consequentially, countability is not 100%, and the birder can choose among the plugins on the basis of the level of bias.
5. In the introduction, the papers cited seem rather random. First, wildlife/bird population assessment has a huge literature, and many books or review papers are available to briefly or thoroughly present the topic. Second Table 1 is an ad hoc summary of the methods. The chosen papers/books are not representative. At the end of the manuscript, the bibliographic information for 4 books is given.
6. Meanwhile, the definitions of the 5 plugins are long enough, but the statistical methods are neglected. No statistical text is referenced!
7. In the text and on the sides of the pages, I left comments. Overall, the text can be shortened, and the clarity needs improvement.
8. In the list of references, several papers are partly cited (missing authors, title, etc.).
10. My suggestions: In general, the text can be shortened (introduction) or improved (statistical methods and definitions), and it must be clearly written. A table summarising the results regarding the five plugins would be a great assistance for the readers. The MS needs a major revision.
I uploaded the pdf with my comments.
Sincerely,
a reviewer

Author Response
- The MS compares five plugins of ImageJ, a software developed for the analysis of micro- to macro-objects. Consequently, the MS is a long review of the capacities of the five plugins, and in scientific terms, it has a low level of novelty.
We thank the reviewer for the question, we acknowledged that the five plugins we tested have been previously used in several studies. However, in this study, we would like to highlight the alternative to previously used bird counting methods namely the particle analyzer and find maxima methods by using three methods in terms of watershed, trainable WEKA segmentation, and TrackMate. As previous studies have stated, the former set of methods has limitations in segmenting and differentiating overlapping objects, therefore we hypothesized that the latter set of methods can be used to overcome this problem. Therefore, the novelty of this paper is to conduct systematical comparison for counting performance for bird for the first time.
- This kind of review can help ornithologists choose among the many similar applications. To achieve this, it would be useful to use some pointing system or list the pros and cons of each plugin. In the text, the reader can have difficulties following their presentations and understanding the advantages or disadvantages.
We thank the reviewer for the comment, as suggested, we have added Table 8 in the revised manuscript to show the advantages and disadvantages of each tested method for bird counting in the manuscript, as we have agreed that this table containing the summary should be simpler and easier to read for the reader.
- The introduction of the MS is lengthy, and the importance of birds and general conservation topics are far beyond the scope of this text. A concise introduction focusing on bird population assessments is right enough.
We would like to thank the reviewer for the question, concerning mentioning the importance of birds and general conservation topics. From our knowledge, population assessments are often used as a parameter in general conservation topics in bird-related studies, thus we mentioned it in the manuscript. However, the reviewer also mentioned that the aforementioned parts are lengthy, therefore we have tried to adjust the manuscript as suggested in this revised manuscript. We truly appreciate for your comments.
- The authors use counting, estimation, and monitoring as synonims. In the wildlife conservation (and related statistical) literature, these words have definitions, and are carefully used. This MS is a text about bird counting with the assumption that, on digital pictures, birds can be much better enumerated than with other kinds of methods. Although the theoretical assumption is 100% countability/visibility, the final results show that, depending on the plugin (for the user, each plugin is a blackbox), over- or underestimation is typical. Consequentially, countability is not 100%, and the birder can choose among the plugins on the basis of the level of bias.
Thank you for the question, Firstly, addressing the reviewer's concern on the use of counting, estimation, and monitoring in the manuscript. The writer agreed there is some misuse of the aforementioned terms in the manuscript. We have adjusted the manuscript as suggested to avoid misinterpretation by the reader, additionally, we would like to add a clarification on the use of these terms in our manuscript.
- The use of counting in the manuscript is to count the number of birds as it is available in the image.
- Estimation is used to describe the inability to count the full population therefore, count results are used to estimate the total population.
- Lastly monitoring is seldom used, it is used to show that researchers usually use “counting” to “estimate” the number of birds to observe their condition.
The concern of less than 100% countability/visibility also exists due to the amount of overlapping/connecting between objects (birds) which we have mentioned in the manuscript. The tools we tested here are present in ImageJ a free-to-use open-source image-processing program widely used by researchers. Although we hypothesized TrackMate to perform the best out of the five tested methods, we also expect it to not work perfectly (100% countability), as even more advanced AI/Machine-learning tools usually only reach a maximum of 99~% countability. We also informed the advantages and disadvantages of each plugin in the revised manuscript as a reference for future users, depending on their biases as the reviewer mentioned.
- In the introduction, the papers cited seem rather random. First, wildlife/bird population assessment has a huge literature, and many books or review papers are available to briefly or thoroughly present the topic. Second Table 1 is an ad hoc summary of the methods. The chosen papers/books are not representative. At the end of the manuscript, the bibliographic information for 4 books is given.
We would like to thank the reviewer for the suggestion. We have adjusted the cited papers in the revised manuscript to better represent the bird population assessment. In addition, we also have made adjustment in Table 1 citations to represent the methods better. We also appreciate the references provided by the reviewer and added them to the manuscript, however as we have limited access to books, we are unable to access Seber, GAF’s publication, therefore we did not add it to the manuscript, we hope the reviewer is able to understand the circumstance.
- Meanwhile, the definitions of the 5 plugins are long enough, but the statistical methods are neglected. No statistical text is referenced!
We would like to appreciate the reminder given by the reviewer, As suggested we have added the explanation and reference for the statistical method we used in this study (Deming linear regression), we also added the explanation for sensitivity, precision, and F1 score, metrics that we used to further identify the best counting method in the revised manuscript.
- In the text and on the sides of the pages, I left comments. Overall, the text can be shortened, and the clarity needs improvement.
We appreciated the reviewer's comments provided in the manuscript, as suggested, we have adjusted the text according to the comments, and we have also tried shortening several parts of the manuscript, especially in the introduction part as mentioned at the start of the reviewer report form.
- In the list of references, several papers are partly cited (missing authors, title, etc.).
We appreciated the comment of the reviewer, as mentioned we acknowledged that there are several references with missing information, therefore we have added the missing information to the references.
- My suggestions: In general, the text can be shortened (introduction) or improved (statistical methods and definitions), and it must be clearly written. A table summarising the results regarding the five plugins would be a great assistance for the readers. The MS needs a major revision.
We thank the reviewer for the overall review. As the reviewer has commented, suggested, and provided a pdf to revise the manuscript, we have taken the suggestions to improve the manuscript further. As suggested, we have tried to shorten the introduction as we have found some redundant sentences and improve the material & methods as we have not explained the statistical methods and the definitions of precision, sensitivity, and F1 score in the earlier version of the manuscript. We also added a Table 8 on summarizing the advantages and disadvantages of each tested method to make it easier for the readers.
Reviewer 2 Report
Comments and Suggestions for Authors
The paper compares five bird counting methods on different data sets and evaluates their performance. The study is well done, but a few more details can be added:
1. What are the underlying algorithms of the tools compared? These could be described, briefly, if known.
2. How can be dealt with systematic errors? Some methods, for example particle analyzer, seem to detect fewer birds compared to ground truth in all data sets. When fine tuning these numbers, the performance may increase.
3. Would it be possible to combine different methods, for example for selection and validation? This could exploit some synergy, e.g. maintaining a precision that no single algorithm can offer.
All in all an interesting study.
Comments on the Quality of English Language
Typos:
l. 26: "different bird number(s) or flying pattern(s)" (use plural)
table 1 is not referred to in the text.
Author Response
The paper compares five bird counting methods on different data sets and evaluates their performance. The study is well done, but a few more details can be added:
- What are the underlying algorithms of the tools compared? These could be described, briefly, if known.
We thank the reviewer for the question. We have tried to briefly describe the tools used in this study in the revised manuscript. First, we would like to address the particle analyzer, the tool is quite simple, users just need to set a color threshold to highlight the objects, then use the Analyze Particles tool available in ImageJ to count the cells by setting the size threshold range depending on the birds' size. The find maxima method uses a Prominence threshold, it will detect the objects by comparing the intensity to the background depending on the prominence threshold. Third is watershed method using an algorithm to separate overlapping/connecting objects, commonly used in cells. The specific algorithm is unknown for watershed method, but the final result is mostly circular/oval shaped. The fourth tested method is Trainable WEKA segmentation, this method uses a collection of machine learning algorithm with selected images to create a model for detection. Trainable WEKA segmentation has a lot of filters implemented in the tools. In this study, we used Gaussian blur, Sobel filter, Hessian, Difference of Gaussian, and Membrane projections were used to train the model. Lastly, TrackMate have several different detectors available to choose. In this study, we used Difference of Gaussian (DoG) detector, which is a faster detector compared to the Laplacian of Gaussian and showed respectable result.
- How can be dealt with systematic errors? Some methods, for example particle analyzer, seem to detect fewer birds compared to ground truth in all data sets. When fine tuning these numbers, the performance may increase.
We would like to thank the reviewer for the question. As we have mentioned in the manuscript, the most common problem for counting is connecting/overlapping objects. Compared to human perception which is able to count these objects differently from the normal ones, the tools we provided have limited ability to distinguish them, especially particle analyzer and find maxima method, which are the simpler method in this study. Therefore, we also used the segmentation methods (watershed and Trainable WEKA segmentation) to overcome this problem. However, the result of the watershed method seems to result in overcounting the number of birds due to the algorithm being not suitable for birds. While the segmentation result of trainable WEKA segmentation sometimes increases the size of the birds, creating a pseudo-connection thus resulting in a worse result in some cases. The final method we tested is TrackMate which used an algorithm to assign identities to detected birds and follow the movement of the birds for better results. However, the results are also not perfect. Compared to other methods, it has relatively high precision and sensitivity in tested cases. In this study, we have tried our best to fine-tune some parameters to achieve the best results, however as we have reported none of the methods works perfectly. More advanced AI/Machine-learning tools can be tested in the future studies to overcome this problem.
- Would it be possible to combine different methods, for example for selection and validation? This could exploit some synergy, e.g. maintaining a precision that no single algorithm can offer.
We would like to thank the reviewer for the question, regarding combining different methods to improve the counting results. We think that with the methods we tested, there is a very slim chance that it will work. Some of the methods, namely particle analyzer, find maxima, and watershed are similar in workflow and exported result, therefore combining these methods might result in similar results or the result will be similar to watershed (overcounting). Trainable WEKA segmentation as we have mentioned in the previous answer in the rebuttal letter, creates a pseudo-connection between birds, therefore if used in combination with the other method will most likely undercount the final counting results. Lastly, TrackMate is a tracking tool, their high precision and sensitivity might be attributed to the algorithm related to tracking. In conclusion, the method we tested might not work in combination, but other tools such as machine learning/AI might be interesting to improve the counting result in the future study.
All in all an interesting study.
Comments on the Quality of English Language
Typos: l. 26: "different bird number(s) or flying pattern(s)" (use plural)
table 1 is not referred to in the text.
Thank you for your comments, we have fixed those problems in the revised manuscript.
Round 2
Reviewer 1 Report
Comments and Suggestions for Authors
I appreciate your efforts to improve the ms. The definition of monitoring [18 in your references] is: "Monitoring, in its most general sense, implies a repeated assessment of status of some quantity, attribute, or task within a defined area over a specified time period. Implied in this definition is the goal of detecting important changes in status of the quantity, attribute, or task." These three criteria make it different from counting, estimation, evaluation, etc.
Author Response
I appreciate your efforts to improve the ms. The definition of monitoring [18 in your references] is: "Monitoring, in its most general sense, implies a repeated assessment of status of some quantity, attribute, or task within a defined area over a specified time period. Implied in this definition is the goal of detecting important changes in status of the quantity, attribute, or task." These three criteria make it different from counting, estimation, evaluation, etc.
We would like to thank the reviewer for the comments, we agree with the reviewer's comments regarding the definition of monitoring as provided by Thompson et al., (reference 18) in the manuscript and its difference from counting, estimation, evaluation, etc. The word monitor/monitoring in the manuscript, line 33, also implies its use to assess the status of wildlife through observation and study of their population to reach scientific understanding, In lines 44 and 45 the monitoring is done to identify the environmental changes/issues, finally in line 68, it is used to show point counting as one of the means for monitoring.
We have also changed the abstract (line 18) and the table 1 title to “monitoring”, rather than “counting” to suit the definition better. Our sentences in the previous reply might not be clear enough, but what we wanted to imply was counting, evaluation, estimation, etc. is a method to do the monitoring as also implied in reference 18.